# Utilization of Lemon Peel for the Production of Vinegar by a Combination of Alcoholic and Acetic Fermentations

**DOI:** 10.3390/foods12132488

**Published:** 2023-06-26

**Authors:** Qingyuan Ou, Jian Zhao, Yuheng Sun, Yu Zhao, Baoshan Zhang

**Affiliations:** College of Food Engineering and Nutritional Science, Shaanxi Normal University, Xi’an 710119, China; oqy18747780946@163.com (Q.O.); jian.zhao@unsw.edu.au (J.Z.); 15129216769@163.com (Y.S.); yuzhao@snnu.edu.cn (Y.Z.)

**Keywords:** lemon peel, fruit vinegar, fermentation, E-nose, E-tongue

## Abstract

Lemon peel is the major by-product of lemon juice processing and is currently underutilized. In this study, we explored the feasibility of using lemon peel as a raw material for making vinegar. Lemon peel was homogenized, treated with pectinase (30,000 U/g, 0.1%) at 50 °C for 4 h, and then filtered. The obtained lemon peel juice was first subjected to alcoholic fermentation by *Saccharomyces cerevisiae* var. FX10, and then acetic fermentation by an acid tolerant *Acetobacter malorum*, OQY-1, which was isolated from the lemon peels. The juice yield of the lemon peel was 62.5%. The alcoholic fermentation yielded a lemon peel wine with an alcoholic content of 5.16%, and the acetic acid fermentation produced a vinegar with a total acid content of 5.04 g/100 mL. A total of 36 volatile compounds were identified from the lemon vinegar, with some compounds such as esters and some alcohols that increased significantly during alcoholic fermentation while alcohols, terpenoids, and some esters decreased significantly during the fermentations. E-nose and E-tongue analyses coupled with principal component and discriminant factor analyses (PCA and DFA) were able to discriminate the samples at different fermentation stages. Overall, this work demonstrates the potential to transform lemon peel into a valuable product, thus reducing the waste of lemon processing and adding value to the industry.

## 1. Introduction

Lemon is a major fruit with an estimated production of 19.7 million metric tons in 2019, which was the 15th most produced fruit globally [1]. Approximately half of lemon production is used for making lemon juice, with a juice extraction rate of about 50% [2], which leaves a huge amount of lemon peel as a waste or by-product. Although a small amount of lemon peel is used to extract essential oils, flavors, and polysaccharides (mainly pectin), the vast majority remains unutilized and discarded as waste. This practice is not only environmentally undesirable, but also throws away a valuable by-product that is rich in nutrients, bioactive compounds, and other useful chemicals. Therefore, it is both economically and environmentally imperative to find ways to utilize lemon peel.

Lemon peel is rich in sugars, vitamins, minerals, and aromatic compounds [3], which makes it an ideal material for transformation by fermentation into products such as vinegar. Vinegar is one of the most common condiments in many countries. Currently, vinegar is mainly produced from the fermentation of grains such as rice and sorghum, malt and grape juice. Small amounts of fruit vinegar are also produced including lemon vinegar, but these are made from fruit juices, rather than from the utilization of fruit peels. There has been scant reporting of using lemon peel to make vinegar. Furthermore, lemon peel is not only rich in fermentable nutrients such as sugar, but it also contains a wide variety of bioactive compounds including phenolic compounds [4,5,6]. These components exhibit strong antioxidant activity and have been shown to confer a range of health benefits such as the regulation of lipid metabolism, the prevention of various cancers, cardiovascular diseases, obesity, protection of the liver, control of blood pressure and sugar, and resistance to fatigue [7,8,9,10]. Therefore, compared with other types of vinegar, fruit vinegar such as lemon vinegar can not only provide the usual sensory functional value as a condiment, but can also offer additional health benefits, potentially making it superior to the traditional types of vinegar.

The objective of this study was to explore the potential of utilizing lemon peel to make a vinegar product by sequential alcoholic and acetic acid fermentations. The alcoholic fermentation was conducted by *Saccharomyces cerevisiae* using lemon juice extracted from lemon peel, while acetic acid fermentation was conducted by an *Acetobacter malorum* strain isolated from the lemon peels. The chemical and sensory characteristics of the lemon peel vinegar were evaluated by solid-phase microextraction (HS-SPME) coupled with gas chromatography-mass spectrometry (GC-MS) as well as an electronic-nose and -tongue (E-nose and E-tongue). 

## 2. Materials and Methods

### 2.1. Extraction of Juice from Lemon Peel 

Lemon (*Citrus limon*) peel was supplied by Dian Company (Tongnan District, Chongqing, China) in sound hygienic conditions. The lemon peel was first homogenized, heated to 50 °C, then added with pectinase (30,000 U/g) at the ratio of 0.1% (*v*/*v*). The enzymatic digestion of the lemon peel homogenate was conducted at 50 °C for 4 h and the mixture was filtered to yield the lemon peel juice with a yield of 62.5% (*w*/*w*). The initial physicochemical properties of the juice were: solid content, 10 °Brix of sucrose equivalent; density, 1.028 g/mL (at 20 °C); pH 2.78. 

### 2.2. Alcoholic Fermentation of the Lemon Peel Juice 

Laboratory-scale alcoholic fermentations were carried out using a 500 mL glass conical flask containing 400 mL of the lemon peel juice with the solid content adjusted to 15 °Bx by the addition of sucrose. A commercial strain of *Saccharomyces cerevisiae* (FX10, LAFFORT, France), preserved in yeast extract agar slant at 4 °C, was activated by subculturing in sterilized lemon peel juice, prepared as described above, and incubating at 30 °C for 24 h to prepare the inoculum. The yeast population in the inoculum was about 10^7^ cfu/mL, which was inoculated into the lemon peel juice at a ratio of 3% (*v*/*v*). The fermentation was conducted at an incubator set at 30 °C, and was ended when the Brix value reached about 8.0°Bx, and the lemon wine thus obtained was stored at 4 °C until further use.

### 2.3. Making of Lemon Vinegar

#### 2.3.1. Isolation and Identification of Acid Tolerant *Acetobacter* spp.

The acid tolerant *Acetobacter* was isolated from lemon peels by streaking the juice sample onto the agar plate of the GYC medium (1% yeast extract, 1% glucose, 2% CaCO_3_, 2% agar *w*/*v*, 4% anhydrous ethanol) prepared by the addition of the lemon peel juice at 85% (*v*/*v*), which was incubated at 30 °C under aerobic conditions for 72 h. After incubation, Gram staining was performed on isolated colonies, and suspect Acetobacter colonies were selected for genomic analysis for identification. Genomic DNA of the strain was extracted and the 16S rDNA gene was cloned with the forward primer 5′-AGAGTTTGATCCTGGCTCAG-3′ and the reverse primer 5′-TACGGCTACCTTGTTACGACTT-3. Nucleotide sequencing of the amplified DNA fragments was performed by Major Biotech Co. Ltd. (Shanghai, China). Using the BLAST database on the NCBI website, the strain’s 16S rDNA gene sequence was compared, and MEGA 7.0 software was used to create a phylogenetic tree to confirm the strain. The identified Acetobacter strain was stored at −20 °C until being used.

#### 2.3.2. Acetic Fermentation of the Lemon Peel Wine

The Acetobacter strain was first activated in 100 mL of nutrient broth incubated at 32 °C for 48 h with rotary shaking at 150 rpm. The bacterial suspension was adjusted to about 10^6^ cfu/mL with sterile distilled water to obtain the inoculum, 25 mL of which was inoculated into 100 mL of the previously prepared lemon wine in 250 mL sterile flasks. The acetic acid fermentation was carried out at 32 °C with rotary shaking at 150 rpm. The fermentation was completed when the alcohol content reached below 1.0% (*v*/*v*). The resulting lemon vinegar was stored at 4 °C until analysis.

### 2.4. Determination of Quality Parameters of the Lemon Vinegar 

The pH of the vinegar was measured using a potentiometer/pH analyzer (ST2100, OHAUS Instruments, Inc., Parsippany, NJ, USA). Titratable acid (TA) was determined by titration with 0.01 M NaOH and expressed as g L^−1^ of citric acid. All of the measurements were performed in triplicate (*n* = 3). The total soluble solids (°Brix) was analyzed using a digital refractometer (Lichen Instrument Technology Co., Shanghai, China). Alcohol content was determined by gas chromatography (SHIMADZU-2010Plus, Japan). Color was measured by a handheld colorimeter (NS810, Guangdong 3nh Intelligent Technology Co., Guangzhou, Guangdong, China), which was calibrated against a blackboard and whiteboard supplied with the instrument. The lightness (L*), green/red component (a*), and blue/yellow component (b*) of the samples were recorded. All of the measurements were performed in triplicate (*n* = 3). 

### 2.5. Analysis of Organic Acids

Organic acids were analyzed by high performance liquid chromatography (HPLC) equipped with a low-pressure quadruplex pump, autosampler, column temperature chamber, and diode array detector (Thermo Fisher, Bremen, Germany). The column used was a XSelect HSS T3 column (100 Å, 5-μm particle size, 4.6 mm × 250 mm, Waters, Milford, MA, USA) with detection at 210 nm [11]. After filtration through a 0.22 μm organic Nylon 66 membrane filter (Jinlong, Tianjin, China), 20 μL of the samples was injected into the HPLC apparatus, and elution was performed isocratically with a mixture of 0.04 mol/L potassium dihydrogen phosphate (pH adjusted to 2.7 by phosphoric acid) and acetonitrile (96.5:3.5 *v*/*v*) as the mobile phase at a flow rate of 0.4 mL/min. Peaks were identified through a comparison of their retention times with standard organic acids analyzed under the same conditions. Quantification was carried out using the external standard method with standard curves constructed from organic acids. 

### 2.6. HS-SPME-GC-MS Analysis of Volatile Components

Volatile components of the samples were analyzed by headspace solid-phase microextraction (HS-SPME) coupled with gas chromatography-mass spectrometry (GC-MS) following the method described by Sun et al. [12] with some modifications. Samples (5 mL) were transferred to 20 mL headspace vials, each with 1.5 g NaCl to facilitate the evaporation of volatiles into the headspace, and 60 μL of an internal standard of 2-octanol (0.32 g/L diluted in methanol). Each vial was immediately sealed with Parafilm, and the samples were equilibrated for 30 min at 50 °C, and subsequently extracted for another 30 min at the same temperature using a 50/30 μm DVB/CAR/PDMS fiber (Supelco, Bellefonte, PA, USA) for 30 min. The fiber was then desorbed directly in the injector of a GC-MS 8890-5977B system (Agilent Technologies, Santa Clara, CA, USA) coupled with a DB-WAX column (30 m × 250 μm × 0.5 μm, Agilent, USA) at 240 °C for 3 min. Chromatographic conditions were programmed as follows: initial column oven temperature was set to 40 °C and maintained for 1 min, increasing at 3 °C/min for 180 °C and maintained there for 5 min, then increasing at 20 °C/min to 230 °C and kept there for 10 min. The carrier gas was helium with a flow rate of 1.0 mL/min. The electron impact (EI) ionization energy was set to 70 eV, and the temperature of the ion source and quadrupole temperature at 230 °C and 150 °C, respectively. The mass scanning range was selected from 40 *m*/*z* to 400 *m*/*z* [13,14]. Compounds were identified based on matching MS spectra with those in the NIST MS Search 2.3 library. The quantification was conducted by the internal standard methods using 2-octanol as the internal standard. The quantity of compounds was calculated as follows: 

Amount of volatile compound (µg/L) = peak area of compound × concentration of internal standard/peak area of internal standard. 

The relative concentrations of volatile compounds in the samples were calculated by comparing the peak area of each compound with that of the internal standard, assuming equal responses for all compounds.

### 2.7. E-Nose Analysis of the Lemon Vinegar

Electronic-nose analysis was performed using a SuperNose (SuperNose, Isenso, Torrance, CA, USA). The E-nose consisted of a sampling apparatus and a detector unit containing an array of 14 interactive sensors made of metal oxides for data recording and preprocessing. Principal component analysis (PCA), linear discriminant analysis (LDA), and radar map analysis were performed using Origin Pro 9.0. For the E-nose analysis, each sample (15 mL) was placed in an airtight vial and covered by sealing films. The volatiles of the sample were pumped into the sensor chamber through an inserted needle. The analysis run was controlled by the equipped software, and the sampling process lasted for 2–3 min and ended automatically. Each sample was taken in quadruplicate and the average of the sensor data was used for subsequent statistical analysis. 

### 2.8. E-Tongue Analysis of the Lemon Vinegar

The electronic-tongue (E-tongue) (Shanghai Ruifen International Trading Co. Ltd, Shanghai, China), also known as a taste sensor or taste fingerprint analyzer, was used to analyze the lemon vinegar following the method described by Li et al. [15] with minor modifications. The E-tongue consisted of an array of six non-specific inert metal sensors with electrodes made of platinum, gold, titanium, palladium, and silver. Vinegar samples (15 mL) were brought to room temperature (20–23 °C). The sensor array of the E-tongue was first immersed into the reference solution (0.01 mol/L potassium chloride) for preheating, and then two samples with the greatest difference were selected for pretesting to ensure the sensor response value was in the range of 0.1–10. The program was set up for automatic sample acquisition, and after the pretest was completed, the sensor array was immersed into sample solutions to collect the sensor response values at equilibrium. Before each new measurement, the sensor array was automatically rinsed in a cleaning solution. When the sample collection was completed, the average response values obtained from each of the six sensors were recorded. PCA and discriminant factor analysis (DFA) were performed using the equipped software. Before data collection, the E-tongue system was activated and calibrated to ensure the reliability of the data.

### 2.9. Statistical Analysis

All analyses were conducted in triplicate. All of the data were presented as the means ± standard deviation of three independent analyses (*n* = 3). One-way analysis of variance (ANOVA) was applied to analyze the differences between means of data (*p* < 0.05) and Tukey’s tests were used to separate means of significant differences. For E-nose and E-tongue data, principal component analysis (PCA) and discriminant function analysis (DFA) were conducted using the WinMuster software. For the GC-MS data, PCA was performed to differentiate samples using SIMCA (version 14.1, Umeå, Sweden). Correlation analysis between E-nose and HS-SPME-GC-MS data was carried out using the Corrplot package in R (version 4.2.3, Auckland, New Zealand). Additional data processing was performed using the Office 2019, SPSS Statistics version 24.0 and Origin Pro 9.0 software package.

## 3. Results and Discussion

### 3.1. Isolation and Identification of Acid Tolerant Acetobacter

Among the strains isolated from the lemon peels, strain OQY-1 displayed the best ability of acid tolerance. OQY-1 was Gram-negative, rod-like, and formed short-chains (Figure 1A). Its colonies were beige in color, moist, irregular in shape, and had a clear halo on acetic acid bacteria agar containing calcium carbonate (Figure 1B). The complete sequence of the 16S rDNA of OQY-1 was 1423 bp long. The species was initially determined by the BLAST program on NCBI (http://www.ncbi.nlm.nih.gov/ (accessed on 3 June 2023)) with more than a 99% similarity of type strains. Then, a phylogenetic tree including the strain OQY-1 and reference strains of each species was constructed with the neighbor-joining method (Figure 1C), showing that OQY-1 was mostly associated with the 16S rDNA of *Acetobacter malorum* strain EW-m (homology > 99%). Combined with the results of the cell and colony morphology observations, the strain OQY-1 was identified as a strain of *Acetobacter malorum*, named as *Acetobacter malorum*, OQY-1.

### 3.2. Alcoholic and Acetic Acid Fermentations

The major physicochemical parameters of the lemon peel juice, wine, and vinegar are shown in Table 1. The alcoholic fermentation of the lemon peel juice was completed in 10 days when the ethanol content reached approximately 5%. The acetic fermentation was completed 28 days after inoculation, with a final acidity of about 5.04 g/100 mL. Both fermentations were relatively inefficient compared with the fermentations of fruit juices. This is likely to be due to the presence of high levels of essential oil components in the lemon peel, which might exert an antimicrobial effect [16]. 

The initial pH of the lemon juice was 2.78, which decreased to 2.46 in the wine and further to 2.01 in the vinegar. Concurrently, the titratable acids increased from 3.43 to 5.04 g/L. These results are in agreement with previous findings [17]. The soluble solids in the lemon peel juice was 15.0 °Brix (adjusted by addition of sucrose), which decreased to 8.0 °Brix in the wine and further to 6.5 °Brix in the vinegar. The major soluble solids of the lemon peel juice were sugars, which, as expected, were gradually utilized by the yeast and acetic acid bacteria during the fermentations.

Fermentation also led to color changes in the products. The L* values declined with the fermentations, indicating that the color of the wine and vinegar was darker than the original juice. The a* value decreased from 1.83 to 1.25, while the b* value increased from 0.40 to 0.90 in the lemon juice and vinegar, respectively. These indicated that with fermentation, the product became more reddish and less yellow in the color hue. Visual observation confirmed that during the fermentation process of lemon juice, the color of the samples changed from yellow to yellowish-brown, which was most likely due to enzymatic browning reactions that occurred to constituents of the lemon juice such as polyphenols.

### 3.3. Organic Acids

Organic acids in the lemon peel juice, wine, and vinegar were analyzed by HPLC and the results are shown in Table 2. As expected, citric acid was the most predominant acid in lemon juice by far, with a level of 6.83 ± 0.02 g/L, which represented over 90% of the total organic acids in the lemon peel juice. Several other organic acids including oxalic, tartaric, and succinic acids were also present in the juice, but in lower concentrations than citric acid. The level of citric and tartaric acids increased during alcoholic fermentation, which is not surprising as many yeasts including *Saccharomyces* spp. are known to produce these organic acids [18]. Furthermore, many bacteria are also able to produce these organic acids [19]. The levels of these organic acids decreased significantly, and in the case of citric acid, it dropped to a non-detectable level in the vinegar. This is expected as acetic acid bacteria including *Acetobacter* spp. are well-known for their ability to utilize organic acids as a carbon source [20].

### 3.4. Volatile Components

Volatile components in the lemon peel juice, wine, and vinegar were analyzed by GC-MS, and the results are shown in Table 3. A total of 84 volatile compounds were identified from the lemon peel juice and its fermentation samples including 30 alcohols, two aldehydes, two acids, nine esters, seven ketones, six phenols, 16 terpene compounds, and 12 other compounds. The relative large number of volatile compounds were expected as *Lemon flavedo* peel is known to be rich in essential oils and has a rich and complex aromatic profile. D-Limonene was the most abundant volatile in the lemon juice with a concentration of 825.88 ± 41.61 mg/L, followed by α-terpineol (552.03 ± 7.42 mg/L), terpinen-4-ol (496.69 ± 7.66 mg/L), and γ-terpinene (416.00 ± 8.25 mg/L). These results are consistent with the literature [21].

Fermentations had a significant impact on the composition of the volatile components. While the concentrations of some components decreased or disappeared altogether, those of some other components increased and new compounds emerged. For example, the concentration of terpinen-4-ol decreased from 496.69 ± 7.6 mg/L in the juice to 56.09 ± 2.94 mg/L in the wine and 44.83 ± 5.78 mg/L in the vinegar, while that of D-limonene dropped to non-detectible levels in the wine and vinegar samples. PCA of the GC-MS data confirmed the trends as the volatile components of the samples were well-separated into three clusters (Figure 2).

Among the aroma compounds found in the lemon peel juice and its fermentation products, terpenoids such as D-limonene, γ-terpinene, β-myrcene, terpinolene, and β-bisabolene were some of the most important aromatic components that contribute to the characteristic fragrance of the products. D-limonene, which was the most abundant volatile compound, was not only a key contributor to the aroma of lemons, but also a “lifting agent” for other volatile compounds [22]. γ-Terpinene was the second most abundant volatile compound in the terpenoids, and is described as “woody”, “tropical”, and “herbal”, while β-myrcene, terpinolene, and β-bisabolene contribute to the “fruity”, “citric”, and “balsamic” aroma [23]. Several minor components are also important contributors to the overall aromatic profile of lemon juice. Linalool has a “fruity” and “floral” note, which is considered an important contributor to the characteristic aroma of citrus fruits [24]. Geraniol, described as “fruity”, “sweet”, “waxy”, and “citrus”, is another important odorant in lemon juice and its fermentation products [25]. Meanwhile, nonanal has a rose, peely, and waxy aroma, while 2-ethylidene-6-methyl-3,5-heptadienal has a fresh, green, and almond note. The concentrations of these compounds were reduced significantly by fermentation. Similarly, the concentrations of non-ethanol alcohols such as linalool, α-terpineol, terpinen-4-ol, geraniol, and some aldehydes such as nonanal and 2-ethylidene-6-methyl-3,5-heptadienal were also reduced during the fermentation. Volatilization was likely the most important reason for the decreases in the concentrations of these volatiles.

On the other hand, some esters such as 6-nonynoic acid, methyl ester, 2,5-octadecadiynoic acid, methyl ester, and bicyclo [3.1.1]hept-2-en-4-ol, 2,6,6-trimethyl-, acetate emerged during fermentation, indicating that they were formed during the fermentation processes.

### 3.5. E-Nose Analysis

The PCA results of the E-nose analysis of the lemon peel juice, wine, and vinegar are shown in Figure 3A. The DI value was 73.18%, indicating that the principal components displayed significant differentiation. Meanwhile, PC1 and PC2 explained 82.86% and 15.47%, respectively, of the total variance for the different fermentation stages of lemon juice, and the first two PCs cumulatively represented 98.33% of the data variance, which provided sufficient information to explain the odor difference of the three different samples. In addition, samples 2 and 3 clustered close together while sample 1 was separated in its own area, indicating that the odor information of samples 2 and 3 was relatively similar, while sample 1 was distant from them. This result further illustrates that there was a significant difference in the odor of the samples before and after fermentation. 

Discriminant function analysis (DFA) was used to classify these three samples (Figure 3B). DFA uses a linear combination of the original variables to construct a discriminant function. The DFA model is based on a search for directions along which the groups are as far apart as possible and the samples of the same group are as close together as possible, which is also used to identify and classify unknown samples. In contrast to PCA, DFA is used to distinguish groups by maximizing the distances between groups, while minimizing the distances between samples in the same group [15]. As can be seen, the individual samples in Figure 3B were distributed in different areas of the graph and did not overlap with each other, with good repeatability. The DI value was 98.85%, indicating that the odors between the three types of samples could be distinguished. The radar plots of the sensor responses of different samples are shown in Figure 3C. The radar maps of wine and vinegar were similar, and both were significantly different from that of the juice. 

### 3.6. Correlation between E-Nose and GC-MS Analyses

Major volatile components identified by GC-MS analysis were used to correlate with the E-nose signals (Figure 4). The signal intensities of the R0 and R7 sensors of the E-nose had positive correlations with the abundances of β-bisabolene, fenchol, γ-terpinene, β-myrcene, δ-terpineol, acetate, terpinolene, terpinen-4-ol, linalyl acetate, D-limonene, nonanal, and α-terpineol, indicating that the sensors were sensitive to terpene, alcohols, some esters, and aldehydes. In contrast, the R0 and R7 signals had a negative correlation with the abundances of isobornyl formate, ethyl acetate, dihydrocarvyl acetate, acetic acid, 3-tetradecanynoic acid, 6-nonynoic acid, and methyl ester, indicating that the sensors were insensitive to acids and most esters. This agreed with the radar plots of the E-nose, which displayed a large difference in the odor signal on the R0 and R7 sensors between the lemon juice, wine, and vinegar samples. In particular, lemon juice produced a larger signal difference when compared with its fermentation products than between the fermentation products. These results demonstrate that the E-nose could discriminate between the aromatic profiles of lemon juice and its fermentation products by responding specifically to different volatile compounds in the samples. 

### 3.7. E-Tongue Analysis

The E-tongue consisted of an array of interaction-sensitive sensors, signal acquisition circuits, and pattern recognition-based data processing algorithms. The PCA results of the E-tongue analysis for the different lemon peel juice products are presented in Figure 5A. PC1 and PC2 explained 66.23% and 16.93%, respectively, of the total variance, indicating that the extracted principal component factors accounted for most of the information from the dataset of the samples. The DI value was 79.78%, suggesting that the PCA was able to effectively distinguish the three samples. Additionally, the data for the three samples were distributed in their own distinct areas without overlapping with each other, further demonstrating that the odor information of the samples were different and could be well-distinguished by the E-tongue. 

Figure 5B shows the results of the discriminant function analysis (DFA) of the samples, with the DI value being 97.35%. It has been reported that if DI reaches 95%, the obtained results can be viewed with high confidence and the analysis method is valid [15]. Thus, the DFA further demonstrated that there were significant differences among the taste of the lemon juice, wine, and vinegar, which could be effectively distinguished by the E-tongue analysis. It should be pointed out that the E-tongue was not used to identify which sample had the superior or favored taste profile, but rather to detect the differences in taste characteristics among the samples. 

In recent years, E-noses and E-tongues coupled with PCA analysis have been used by several researchers to discriminate different types of vinegar. For example, Wang et al. [26] used the technique to successfully distinguish vinegar produced from different brewing techniques. Jo et al. [27] combined an E-nose and E-tongue with MS analysis to differentiate aged vinegar made from different raw materials. However, our study is the first attempt to use the technique for monitoring aroma evolution during vinegar production from lemon peel juice fermentation.

## 4. Conclusions

This study demonstrated that lemon peel is an appropriate raw material for the production of vinegar by sequential alcoholic and acetic acid fermentations by *Saccharomyces cerevisiae* var. FX10 and *Acetobacter malorum*, OQY-1, respectively. The vinegar produced contained 5.04% acetic acid, with a distinct aroma and taste profile that could be distinguished from the lemon juice and lemon wine by the E-nose and E-tongue. A total of 36 volatile compounds were isolated from the lemon vinegar, with some compounds such as esters and some alcohols increasing significantly during alcoholic fermentation while alcohols, terpenoids, and some esters decreasing significantly during acetic acid fermentation. Furthermore, this study showed that E-noses and E-tongues are useful tools for the analysis of the fermentation process and can be used to discriminate the samples at different fermentation stages. Overall, this work demonstrates the potential to transform lemon peel, a by-product of lemon juice processing, into a valuable product, thereby reducing the waste of lemon processing and adding value to the industry, thus making the industry economically and environmentally more sustainable.

## Figures and Tables

**Figure 1 foods-12-02488-f001:**
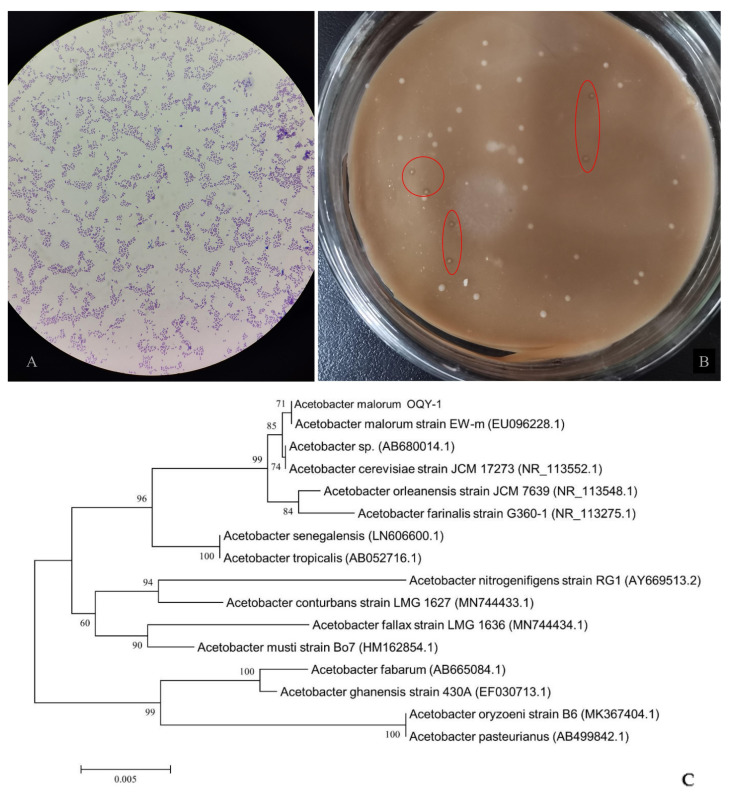
(**A**) Cell morphology of the strain OQY-1 under a microscope; (**B**) colony morphology of the strain OQY-1 on acetic acid agar; (**C**) a phylogenetic tree based on 16S rDNA gene sequences of the strain OQY-1 and other *Acetobacter* species. Percent bootstrap values above 60 (1000 replicates) were indicated at nodes. Scale bar = 0.0050 substitutions per nucleotide position. Sequence alignment and comparison was performed using MEGA version 7.0. Neighbor-joining trees were constructed using MEGA version 7.0.

**Figure 2 foods-12-02488-f002:**
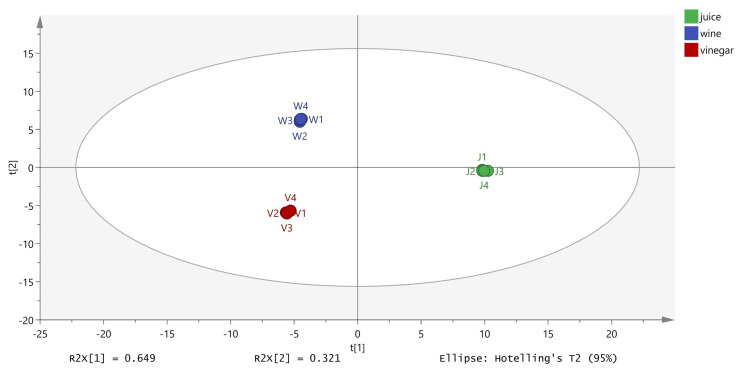
Principal component analysis of volatile compounds (GC-MS data) in lemon juice, wine, and vinegar.

**Figure 3 foods-12-02488-f003:**
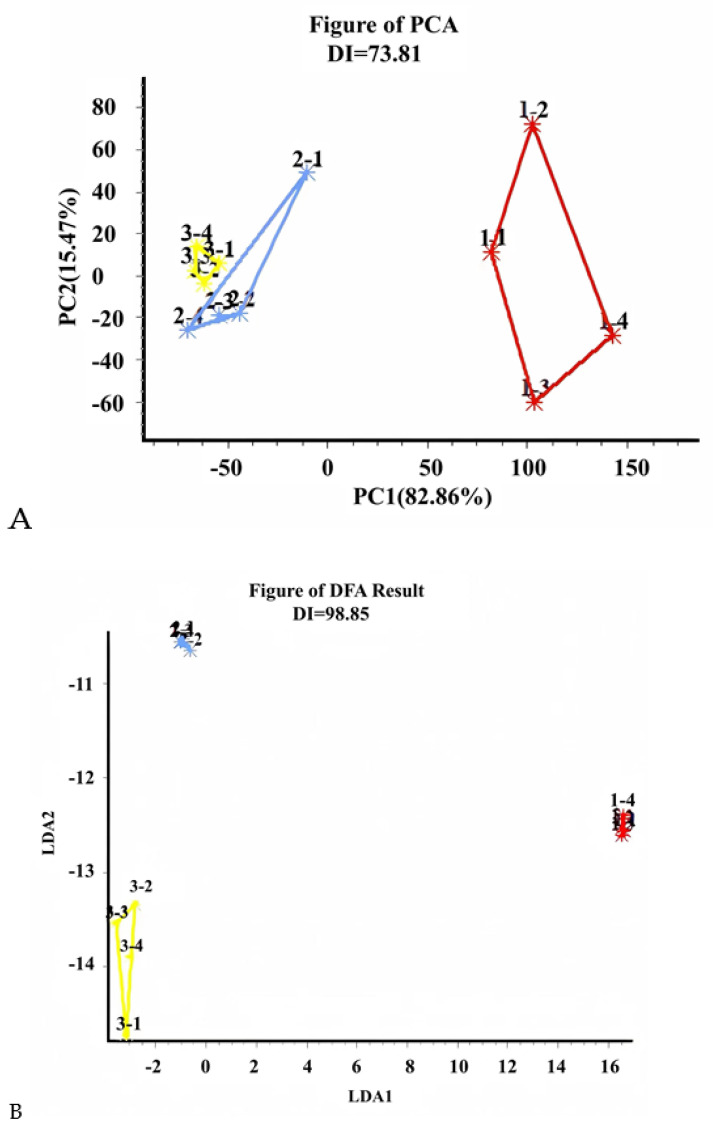
The results of the principal component analysis (PCA) (**A**), discriminant function analysis (DFA) (**B**), and radar map (**C**) of the E-nose analysis of lemon juice, wine, and vinegar. In (**A**,**B**), the red colored cluster indicates the lemon peel juice, the blue colored cluster indicates lemon wine, and the yellow colored cluster indicates lemon vinegar.

**Figure 4 foods-12-02488-f004:**
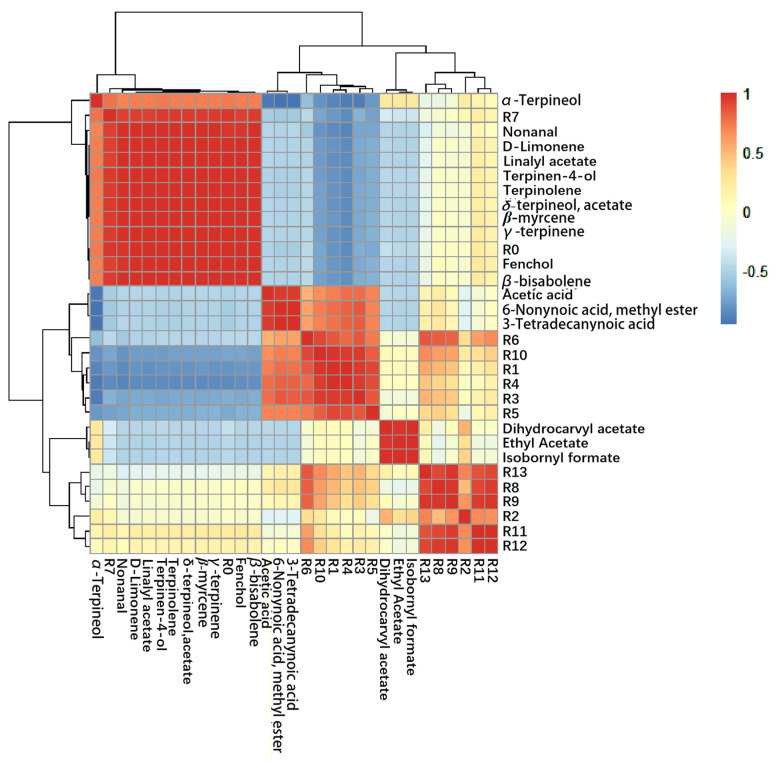
Correlations between the results of the E-nose and GC-MS analysis of the lemon peel juice, wine, and vinegar. The color from blue to red represents correlations ranging from negative to positive relationships.

**Figure 5 foods-12-02488-f005:**
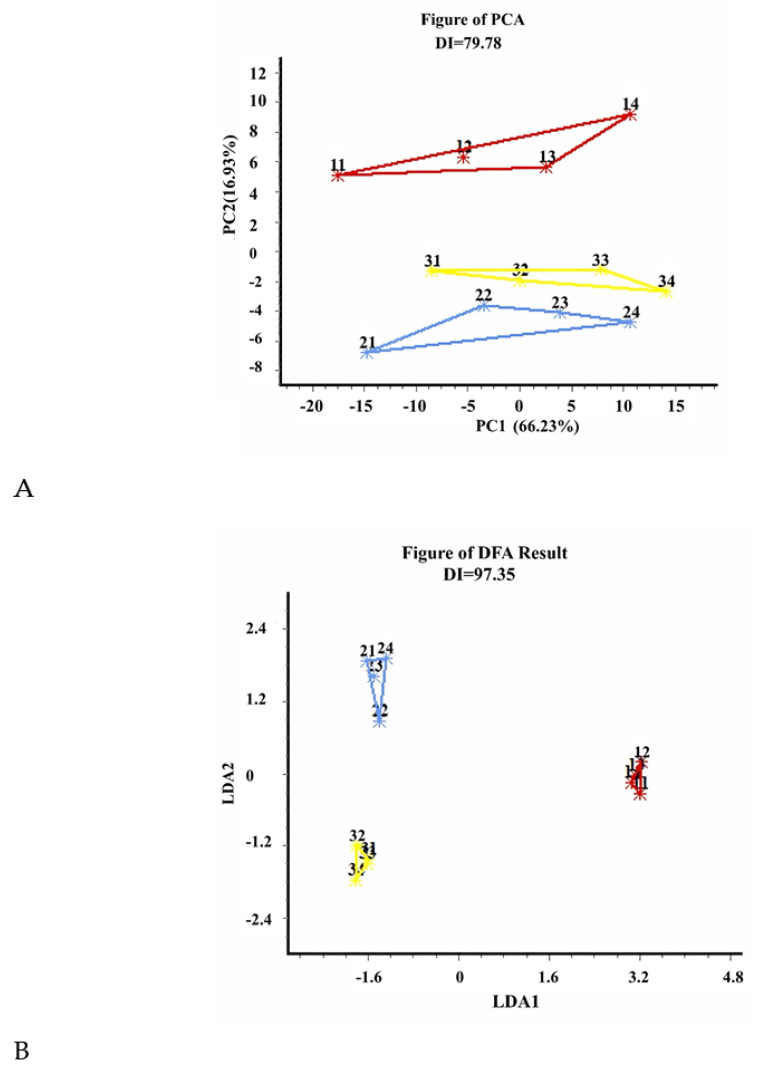
The results of the principal component analysis (PCA) (**A**), discriminant function analysis (DFA) (**B**) of the E-tongue analysis for the discrimination of lemon juice (red colored cluster), wine (blue colored cluster), and vinegar (yellow colored cluster).

**Table 1 foods-12-02488-t001:** Physicochemical properties of lemon peel juice, wine, and vinegar.

	Juice	Wine	Vinegar
pH value	2.78 ± 0.01 ^a^	2.46 ± 0.02 ^b^	2.01 ± 0.02 ^c^
Titratable acidity (g/L)	3.43 ± 0.00 ^c^	4.45 ± 0.01 ^b^	5.04 ± 0.01 ^a^
Ethanol (%*v*/*v*)	n. a. ^c^	5.16 ± 0.02 ^a^	<1.00 ^b^
Chromaticity	L* = 0.23 ± 0.02 ^a^ a* = 1.83 ± 0.01 ^a^ b* = 0.40 ± 0.01 ^c^	L* = 0.14 ± 0.01 ^b^ a* = 1.81 ± 0.02 ^a^ b* = 0.59 ± 0.01 ^b^	L* = 0.11 ± 0.01 ^c^ a* = 1.25 ± 0.00 ^b^ b* = 0.90 ± 0.02 ^a^
TSS (°Brix)	15.0 ± 0.00	8.0 ± 0.00	6.5 ± 0.00

Results are expressed as the mean ± standard deviation (independent samples, *n* = 3). Values in the same rows with different superscript letters differ significantly (*p* < 0.05).

**Table 2 foods-12-02488-t002:** Concentrations of organic acids in lemon juice, wine, and vinegar.

Sample								
	Organic Acids (g/L) ^a^							
	Oxalic	Tartaric	Lactic	Acetic	Citric	Succinic	Fumaric	Trans-Aconitic
Lemon juice	2.56 ± 0.28 ^a^	0.73 ± 0.12 ^b^	n. a. ^c^	n. a. ^b^	6.83 ± 0.02 ^b^	1.76 ± 0.10 ^a^	0.10 ± 0.01 ^a^	0.24 ± 0.01 ^a^
Lemon wine	0.93 ± 0.02 ^b^	3.74 ± 0.10 ^a^	5.19 ± 0.05 ^a^	n. a. ^b^	7.19 ± 0.02 ^a^	0.36 ± 0.02 ^c^	0.11 ± 0.01 ^a^	0.09 ± 0.03 ^b^
Lemon vinegar	0.25 ± 0.01 ^c^	0.46 ± 0.01 ^c^	0.23 ± 0.02 ^b^	184.34 ± 2.73 ^a^	n. a. ^c^	0.62 ± 0.04 ^b^	0.12 ± 0.01 ^a^	0.05 ± 0.01 ^b^

Results are expressed as the mean ± standard deviation (independent samples, *n* = 3). Values in the same rows with different superscript letters differ significantly (*p* < 0.05).

**Table 3 foods-12-02488-t003:** Main volatile compounds identified in different fermentation stages of lemon peel juice.

Peak Number	Compounds	RT (min)	Concentration (mg/L)		
			Lemon Juice	Lemon Wine	Lemon Vinegar
	**Alcohols**	-			
1	Ethanol	3.615	ND ^b^	42.46 ± 1.14 ^a^	ND ^b^
2	Dihydrocarveol	8.213	ND ^b^	1.98 ± 0.17 ^a^	ND ^b^
3	2-methyl-5-(1-methylethenyl)-, (1.alpha.,2.alpha.,5.beta.)-cyclohexanol	11.535	ND ^b^	40.55 ± 0.96 ^a^	ND ^b^
4	trans-3-caren-2-ol	13.956	ND ^b^	ND ^b^	0.55 ± 0.05 ^a^
5	2-hydroxy-2,2,4-trimethyl-3-cyclohexene-1-methanol	16.202	ND ^b^	1.54 ± 0.16 ^a^	1.65 ± 0.13 ^a^
6	2,6,6-trimethyl-bicyclo [3.1.1]heptan-2-ol	17.965	ND ^c^	0.10 ± 0.02 ^b^	0.33 ± 0.04 ^a^
7	Carveol	19.033	5.52 ± 0.10 ^a^	ND ^b^	ND ^b^
8	6-methyl-2-heptanol	21.908	7.12 ± 0.14 ^a^	ND ^b^	ND ^b^
9	4-(1,1-dimethylethyl)-benzenemethanol	24.443	ND ^b^	0.19 ± 0.03 ^a^	ND ^b^
10	Linalool	25.328	26.39 ± 2.28 ^a^	13.57 ± 0.72 ^b^	1.71 ± 0.10 ^c^
11	Para-menth-3-en-1-ol	26.571	ND ^b^	319.50 ± 5.39 ^a^	306.12 ± 12.02 ^a^
12	Fenchol	26.612	169.69 ± 13.47 ^a^	ND ^b^	ND ^b^
13	Terpinen-4-ol	27.359	496.69 ± 7.66 ^a^	56.09 ± 2.94 ^b^	44.83 ± 5.78 ^c^
14	1-methyl-4-(1-methylethenyl)-cyclohexanol	28.383	ND ^c^	17.43 ± 2.50 ^b^	26.17 ± 2.18 ^a^
15	5-isopropyl-2-methylbicyclo [3.1.0]hexan-2-ol	28.48	8.99 ± 0.16 ^a^	ND ^b^	ND ^b^
16	Isoborneol	29.528	0.92 ± 0.10 ^a^	ND ^b^	ND ^b^
17	cis-verbenol	30.308	0.95 ± 0.19 ^a^	ND ^b^	ND ^b^
18	α-terpineol	31.208	552.03 ± 7.42 ^a^	401.51 ± 10.04 ^b^	4.44 ± 0.12 ^c^
19	p-mentha-1,5-dien-8-ol	31.892	56.02 ± 8.12 ^a^	ND ^b^	ND ^b^
20	cis-3,7-dimethyl-2,6-octadienol	32.526	53.67 ± 3.35 ^a^	ND ^b^	ND ^b^
21	Benzyl alcohol	35.718	12.74 ± 0.09 ^a^	ND ^b^	ND ^b^
22	m-methylbenzyl alcohol	35.766	ND ^c^	11.49 ± 0.64 ^a^	7.02 ± 0.70 ^b^
23	Geraniol	36.554	4.02 ± 0.06 ^a^	ND ^b^	ND ^b^
24	Phenylethyl alcohol	37.294	ND ^b^	ND ^b^	23.09 ± 0.96 ^a^
25	2-methyl-benzeneethanol	37.302	ND ^b^	4.77 ± 0.08 ^a^	ND ^b^
26	Caryophyllenyl alcohol	43.512	ND ^c^	3.30 ± 0.15 ^a^	2.91 ± 0.08 ^b^
27	Globulol	48.926	ND ^c^	0.19 ± 0.01 ^b^	0.28 ± 0.03 ^a^
28	α-bisabolol	49.568	0.05 ± 0.03 ^a^	ND ^b^	ND ^b^
29	Hydroxycitronellol	53.174	0.13 ± 0.03 ^a^	ND ^b^	ND ^b^
30	Heptaethylene glycol	59.259	0.14 ± 0.02 ^a^	ND ^b^	ND ^b^
	**Terpenes**	-			
1	(E,E)-1,3,5-heptatriene	4.208	5.19 ± 0.09 ^a^	ND ^b^	ND ^b^
2	α-pinene	5.557	24.66 ± 3.40 ^a^	ND ^b^	ND ^b^
3	Camphene	6.994	0.76 ± 0.12 ^a^	ND ^b^	ND ^b^
4	β-myrcene	10.682	75.96 ± 1.15 ^a^	ND ^b^	ND ^b^
5	(+)-sabinene	10.902	64.09 ± 0.83 ^a^	23.47 ± 3.50 ^b^	4.53 ± 0.15 ^c^
6	D-limonene	11.625	825.88 ± 41.61 ^a^	ND ^b^	ND ^b^
7	(+)-4-carene	13.225	ND ^b^	ND ^b^	7.37 ± 0.23 ^a^
8	γ-terpinene	13.33	416.00 ± 8.25 ^a^	7.00 ± 0.47 ^b^	5.93 ± 0.63 ^b^
9	α-terpinene	14.752	ND ^b^	ND ^b^	9.52 ± 0.11 ^a^
10	Terpinolene	14.769	95.58 ± 0.06 ^a^	10.57 ± 1.36 ^b^	8.21 ± 0.40 ^c^
11	γ-elemene	28.366	19.84 ± 0.77 ^a^	ND ^b^	ND ^b^
12	Limonene oxide	30.235	ND ^b^	24.18 ± 0.41 ^a^	ND ^b^
13	β-bisabolene	33.679	106.20 ± 4.10 ^a^	ND ^b^	ND ^b^
14	4-isopropenyl-1-methoxymethoxymethyl-cyclohexene	34.426	ND ^b^	1.71 ± 0.18 ^a^	ND ^b^
15	R-limonene	34.678	ND ^b^	ND ^b^	0.53 ± 0.09 ^a^
16	Dipentenedioxide	41.814	0.34 ± 0.02 ^b^	0.80 ± 0.48 a^b^	1.28 ± 0.06 ^a^
	**Esters**	-			
1	Ethyl acetate	3.111	ND ^b^	2.06 ± 0.06 ^a^	ND ^b^
2	Dihydrocarvyl acetate	7.027	ND ^b^	1.00 ± 0.19 ^a^	ND ^b^
3	Linalyl acetate	8.294	2.02 ± 0.10 ^a^	ND ^b^	ND ^b^
4	δ-terpineol acetate	12.364	152.77 ± 2.28 ^a^	ND ^b^	ND ^b^
5	Isobornyl formate	29.455	ND ^b^	10.56 ± 0.58 ^a^	ND ^b^
6	6-nonynoic acid, methyl ester	30.194	ND ^b^	ND ^b^	58.95 ± 2.95 ^a^
7	2,5-octadecadiynoic acid, methyl ester	40.389	ND ^b^	ND ^b^	0.07 ± 0.01 ^a^
8	Bicyclo [3.1.1]hept-2-en-4-ol, 2,6,6-trimethyl-, acetate	41.355	ND ^c^	0.08 ± 0.03 ^b^	0.21 ± 0.02 ^a^
9	[1,1’-bicyclopropyl]-2-octanoic acid, 2’-hexyl-, methyl ester	45.855	ND ^b^	1.38 ± 0.06 ^a^	ND ^b^
	**Aldehydes**	-			
1	Nonanal	19.374	11.56 ± 1.64 ^a^	ND ^b^	ND ^b^
2	2-ethylidene-6-methyl-3,5-heptadienal	38.333	0.08 ± 0.02 ^a^	ND ^b^	ND ^b^
	**Ketones**	-			
1	6-camphenone	6.613	ND ^b^	0.53 ± 0.13 ^a^	ND ^b^
2	6-methyl-5-hepten-2-one	16.466	0.82 ± 0.05 ^a^	ND ^b^	ND ^b^
3	Fenchone	19.098	ND ^b^	1.12 ± 0.15 ^a^	ND ^b^
4	(+)-2-bornanone	23.647	5.74 ± 0.05 ^b^	7.14 ± 0.70 ^a^	ND ^c^
5	3-methyl-6-(1-methylethyl)-2-cyclohexen-1-one	31.892	ND ^c^	6.57 ± 0.04 ^a^	4.54 ± 0.30 ^b^
6	1-(2-methylphenyl)-ethanone	32.891	ND ^c^	13.21 ± 0.43 ^a^	11.47 ± 0.99 ^b^
7	1-(4-methylphenyl)-ethanone	32.94	18.97 ± 0.25 ^a^	ND ^b^	ND ^b^
	**Acids**	-			
1	Acetic acid	31.762	ND ^b^	ND ^b^	0.62 ± 0.15 ^a^
2	3-Tetradecanynoic acid	45.506	ND ^b^	ND ^b^	1.94 ± 1.40 ^a^
	**Phenols**	-			
1	Butylated hydroxytoluene	39.048	2.27 ± 0.05 ^a^	2.77 ± 0.89 ^a^	3.26 ± 0.27 ^a^
2	p-Cresol	42.557	ND ^c^	5.29 ± 0.16 ^b^	5.69 ± 0.28 ^a^
3	2-methyl-5-(1-methylethyl)-phenol	46.473	ND ^b^	ND ^b^	3.30 ± 0.07 ^a^
4	Thymol	47.244	3.74 ± 0.13 ^b^	4.89 ± 0.03 ^a^	2.44 ± 0.22 ^c^
5	2,6-bis(1,1-dimethylethyl)-phenol	51.493	ND ^b^	0.65 ± 0.04 ^a^	ND ^b^
6	2,4-Di-tert-butylphenol	51.558	ND ^b^	ND ^b^	0.70 ± 0.07 ^a^
	**Others**	-			
1	Dimethyl ether	3.501	ND ^b^	ND ^b^	73.46 ± 0.51 ^a^
2	1-chloro-2-methyl-propane	7.019	1.70 ± 0.05 ^b^	ND ^c^	2.13 ± 0.15 ^a^
3	6,6-bicyclo [3,1,1]heptane	8.31	90.67 ± 10.45 ^a^	ND ^b^	ND ^b^
4	1-methyl-3-(1-methylethyl)-benzene	13.964	ND ^b^	2.45 ± 0.24 ^a^	ND ^b^
5	o-Cymene	13.988	18.15 ± 0.22 ^a^	ND ^b^	ND ^b^
6	Geranyl vinyl ether	19.285	0.02 ± 0.00 ^c^	1.91 ± 0.04 ^a^	0.73 ± 0.01 ^b^
7	4-ethenyl-1,2-dimethyl-benzene	20.511	172.93 ± 6.31 ^a^	ND ^b^	ND ^b^
8	2-benzoyl-1,3-dithiane	22.875	0.39 ± 0.04 ^a^	0.38 ± 0.02 ^a^	ND ^b^
9	(Z)-4-hexadecen-6-yne	23.939	ND ^b^	ND ^b^	17.93 ± 0.20 ^a^
10	3-caren-10-al	42.614	1.35 ± 0.05 ^a^	ND ^b^	ND ^b^
11	4-hydroxy-2,2,4-trimethyl-cyclohexanemethanol	44.141	ND ^b^	6.18 ± 0.27 ^a^	5.72 ± 1.06 ^a^
12	15-Crown-5	54.092	0.06 ± 0.01 ^a^	ND ^b^	ND ^b^

ND: not detected. Results are expressed as the mean ± standard deviation (independent samples, *n* = 3). Values in the same rows with different superscript letters differ significantly (*p* < 0.05).

## Data Availability

The data used to support the findings of this study can be made available by the corresponding author upon request.

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
