# Peer review of "Utilization of Lemon Peel for the Production of Vinegar by a Combination of Alcoholic and Acetic Fermentations"

_foods, 2023, doi:10.3390/foods12132488_

Round 1

Reviewer 1 Report

The attached manuscript has a number of errors and deficiencies, which are entered in the attached document. It is not clear from what is written in the methods how the experiment was performed; what was done in triplicate? Independent samples (N=3) are mentioned in the results. Some of the results "do not match" or are inconsistent. In addition, the image resolution is poor and/or the legend is missing. The first part of the discussion is (quite) adequate, while the continuation itself (volatile compound analysis, E-nose and E-tongue) is very poor. This part needs to be completed.

Author Response

We thank the reviewer for the constructive comments, and have revised the manuscript accordingly. Specifically:

1) We have clarified the replications of the analyses.

2) We have corrected the "non-matching" results. This is due to our oversight in describing some of the results.

3) We up replaced the images with ones that have improved resolutions.

4). We have added some discussion on the E-nose and E-tongue results.

5) We have corrected all the typo (including spacing) errors in the text as highlighted by the reviewer in the attached file.

Reviewer 2 Report

This paper titled "Utilization of Lemon Peel for the Production of Vinegar: Sequential Alcoholic and Acetic Acid Fermentations" explores the potential of using lemon peel to produce vinegar through a two-step fermentation process. The authors aim to evaluate the chemical and sensory characteristics of the resulting lemon peel vinegar.

The study begins with the extraction of juice from lemon peel, followed by alcoholic fermentation using Saccharomyces cerevisiae and acetic acid fermentation using an isolated strain of Acetobacter malorum. The quality parameters of the lemon vinegar, including pH, titratable acidity, total soluble solids, alcohol content, and color, were determined. The analysis of organic acids and volatile components was performed using high-performance liquid chromatography (HPLC) and headspace solid-phase microextraction coupled with gas chromatography-mass spectrometry (HS-SPME-GC-MS). Electronic nose (E-nose) and electronic tongue (E-tongue) analyses were conducted to assess the sensory characteristics of the vinegar.

The paper provides a detailed methodology and experimental procedures, allowing for replication of the study. However, there are some areas that could benefit from further clarification. The isolation and identification of the Acetobacter strain could be described in more detail, including the specific genomic analysis methods used. Additionally, the authors should provide more information about the E-nose and E-tongue analyses, such as the number and type of sensors used, and the specific parameters and algorithms employed for data analysis.

Overall, the paper presents an interesting investigation into the production of vinegar using lemon peel. The combination of chemical and sensory analyses provides a comprehensive understanding of the characteristics of the lemon peel vinegar. The findings could have potential applications in the food industry, particularly for the development of new vinegar products with enhanced sensory and health benefits.

no issues

Author Response

We thank the reviewer for the encouraging comments, and have revised the manuscript based on the suggestions of the reviewer. Specifically:

1) We have added the types and number of E-nose and E-tongue sensors and the materials used for the manufacture of the sensors. The analysis methods and algorithms have been described in the originally manuscript (PCA, DFA, LDA, and least squares). Additional statistical analysis methods used were described in section 2.9.

2) We believe the description of the genomic analysis methods is sufficiently detailed that can allow repetition of the experiments. Additional information on how the microbial species was identified was provided in Results and Discussion.